# A two-stage random-effects estimator for meta-analyses of the value per statistical life

Stephen C. Newbold[1]*, Chris Dockins[2], Nathalie Simon[2], Kelly Maguire[3], Abdullah Muhammad Sakib[4]

**1** Department of Economics, University of Wyoming, Laramie, Wyoming, United States of America, **2** National Center for Environmental Economics, U.S. Environmental Protection Agency, Washington, D.C., United States of America, **3** Economic Research Service, U.S. Department of Agriculture, Washington, D.C., United States of America, **4** Department of Economics, University of Vermont, Burlington, Vermont, United States of America

* snewbold@uwyo.edu

**Data availability statement:** The data relevant to this study is available from GitHub at https://github.com/scnewbold/2SRE.

## Abstract

We developed and examined the performance of a two-stage random-effects meta-analysis estimator for synthesizing published estimates of the value per statistical life (VSL). The meta-estimation approach accommodates unbalanced panels with one or multiple observations from each independent group of primary estimates, and distinguishes between sampling and non-sampling sources of error, both within and between groups. We used Monte Carlo simulation experiments to test the performance of the meta-estimator on constructed datasets. Simulation results indicate that, when applied to datasets of modest size, the approach performs best when the within-group non-sampling error variances are assumed to be homogeneous among groups. This allows for two levels of non-sampling errors while preserving degrees of freedom and therefore increasing statistical efficiency. Simulation results also show that the estimator compares favorably to several other commonly used meta-analysis estimators, including other two-stage estimators. As a demonstration, we applied the approach to a pre-existing meta-dataset including 113 VSL estimates assembled from 10 revealed preference and 9 stated preference studies conducted in the U.S. and published between 1999 and 2019.

## 1 Introduction

Analysts often use quantitative predictive models to aid in the design and evaluation of public policy interventions, and generally one or more key parameters of such models are not known with certainty. In some domains, many studies have reported multiple competing estimates of an important parameter using more or less credible research methods. In these cases, some means of synthesizing the available estimates—into a single best point estimate, a credible range, or a probability distribution—is needed for use in quantitative policy evaluations.

A leading method for this task is meta-analysis, which is a statistical approach for estimating the central tendency and examining the factors that influence the variation among multiple estimates of an unknown quantity of interest from different studies [1,2]. Meta-analysis

**Funding:** The author(s) received no specific funding for this work.

**Competing interests:** The authors have declared that no competing interests exist.

has been used to synthesize quantitative results from empirical studies in a wide variety of public policy domains, including job search and training programs [3], the impacts of ethanol regulations on corn prices [4], the efficacy of nudges for improving public health [5], the influence of education on intelligence [6], COVID-19 infection fatality rates [7], the value per statistical life [8], and many more.

The "value per statistical life" (VSL) quantifies people's willingness to pay for small reductions in their risk of death [9]. Specifically, the VSL corresponds to the total dollar value associated with a small change in the risk of dying that, when aggregated over a large population, yields one statistical life. For example, if 100,000 individuals are each willing to pay, on average, $100 for a reduction in their risk of dying in the coming year of 1/100,000, then the value of reducing the expected number of deaths in the group by one—i.e., saving one "statistical life"—equals $100,000 \times \$100$, or $10 million [10]. Banzhaf [11] describes the historical origins of the VSL concept, and Cropper et al. [12] provide a broad overview of estimation approaches and applications of the VSL in benefit-cost analysis.

The VSL is among the most important quantities used in benefit-cost analyses of public policies related to health, safety, and the environment as reduced mortality often comprises the largest category of benefits for these actions [12,13]. For example, in the formal regulatory impact analysis of recent revisions to the U.S. Environmental Protection Agency's National Ambient Air Quality Standards, over 98 percent of the monetized benefits were attributed to avoided statistical deaths [14]. The VSL also is commonly used for global public health assessments, including a recent "mortality cost report card" for COVID-19 deaths worldwide [15].

Hundreds of VSL estimates have been reported in the peer-reviewed literature, and more than a dozen previous meta-analyses have been conducted to synthesize multiple estimates of the VSL and examine the factors that influence their magnitudes. However, previous VSL meta-analyses have typically focused on a sub-set of the literature, either hedonic wage or stated preference studies but rarely both. Many also used a single VSL estimate per study or independent data sample, or, when multiple estimates per study were available, these were averaged to produce a single central study estimate before being combined with estimates from other studies. In contrast, the approach we use below accommodates unbalanced panels with one or multiple observations from each independent group of primary estimates and distinguishes between sampling and non-sampling sources of error, both within and between groups. To demonstrate the approach, we use VSL estimates assembled in an earlier U.S. EPA report [16] including observations from both the hedonic wage and stated preference literatures. In this paper we focus on the VSL, but we would expect many of the same data features to characterize meta-analyses in other domains as well, so the estimator performance comparisons and application described here should provide insights with broader implications.

Precise estimates of the VSL play a critical role in informing benefit-cost analyses for policies aimed at improving public health, safety, and environmental quality. For example, in public health, estimates of the VSL can be used to guide investments in vaccination programs by quantifying the monetary benefits of lives saved, ensuring resources are allocated efficiently [17]. In safety regulations, more precise VSL estimates could help set standards for automobile safety features, such as airbags or collision avoidance settings for driverless cars [18], that better balance their costs with the benefits of reducing traffic fatalities [19]. Similarly, in environmental policy, VSL estimates often influence decisions on pollution control measures, such as setting emissions limits for factories [12]. These applications demonstrate how robust meta-analytic methods can bolster evidence-based policymaking and maximize the net benefits of public health and environmental regulations by providing more precise estimates of the key factors that drive policy evaluations.

We make two main contributions in this study. First, we develop a new multilevel meta-analysis estimator and test its performance against other comparable estimators in a Monte Carlo simulation experiment. Second, we demonstrate the use of our new estimator by applying it to a publicly available meta-dataset of estimates of the value per statistical life (VSL). Our estimation approach can deliver higher precision while accommodating a more general error structure than previous VSL meta-analyses.

## 1.1 Previous VSL meta-analyses

While there are several excellent reviews and summaries of the VSL literature [20,21], here we focus on statistical meta-analyses. Most previous VSL meta-analyses have synthesized hedonic wage-based estimates of the VSL. Mrozek and Taylor [8] performed a meta-regression of 203 estimates from 33 hedonic wage studies. Weighted least squares was used for estimation, with weights equal to the inverse of the number of estimates from the parent study, giving each study equal weight. Precision weights were not used because standard errors were not reported in many of the source studies. Viscusi and Aldy [22] used single estimates from each of 44 to 46 studies in six meta-regression model specifications, also without precision weighting. Bellavance et al. [23] used a mixed-effects regression model to combine 39 estimates drawn from 37 hedonic wage studies. Estimates were chosen from each independent data sample (in most cases selecting a single estimate per study) based on similarity of the estimating equation with other studies, the original authors' preferred estimate, and other best-practice considerations. Nelson [24] used the data assembled by Bellavance et al. [23] plus additional hedonic wage observations from the U.S. Environmental Protection Agency [25] in a "tentative and exploratory" meta-analysis of VSL estimates. After dropping outliers, single estimates from 28 primary studies were included in the final meta-dataset for four meta-regression specifications—OLS, fixed-effect, and two versions of random-effects models—which included use of the inverse standard errors as a test for publication bias.

The potential influence of publication bias on reported VSL estimates has been the focus of several meta-analyses of hedonic wage studies, first by Doucouliagos et al. [26] who found significant bias using the Bellavance et al. [23] data set, and later in a series of articles by Viscusi and co-authors. Viscusi [27] constructed a sample of 550 hedonic wage estimates based on 17 studies that used workplace fatality risks calculated from the Census of Fatal Occupational Injury (CFOI) dataset, and compared VSL estimates and publication selection bias in this set to that found in other hedonic wage datasets, including that constructed by Bellavance et al. [23]. Estimates were weighted by inverse variance, and fixed- and random-effects variants of meta-regression models were estimated. CFOI-based estimates exhibited relatively little publication bias. Viscusi and Masterman [28] examined publication bias in U.S. and non-U.S. VSL estimates using a larger international dataset of 1,025 observations from 68 hedonic wage studies. The authors used weighted least squares with inverse variances of the VSL estimates used as observation weights. A quantile regression approach was used to examine publication bias at different levels of VSL estimates. Little evidence of publication bias was found in CFOI-based estimates, but there was evidence of strong bias among non-U.S. studies, which the authors attributed to an anchoring effect of previously published U.S. VSL estimates. Viscusi [30] further examined publication bias in the hedonic wage literature, comparing bias in "best-set" samples (i.e., 1 selected estimate per study) with that found when all study estimates are used. Weighted least squares results suggested that publication bias is statistically significant for both samples but is larger for the best-set sample. The central bias-adjusted VSL estimate for the all-set sample was $8.8 million (2020 U.S. dollars.)

Fewer meta-analyses of stated preference-based VSL estimates have been conducted. Dekker et al. [29] used a Bayesian estimation approach in their meta-analysis of 77 estimates from 26 international contingent valuation studies conducted in 15 countries, with the goal of examining the effect of risk context on VSL estimates. Specifically, they estimated correction factors for "out of context" benefit transfers based on CV studies focused on air pollution, road safety, or those considered "context free." The study by Lindhjem et al. [31] is perhaps the most comprehensive global meta-analysis of stated preference VSL estimates, with 850 estimates drawn from 76 studies conducted in 38 countries. The authors focused on the effects of population characteristics, risk type and context, survey format, and statistical method- ological choices on the VSL estimates. For a subset of the primary studies, the authors com- pared results using alternative weighting schemes—the inverse of the number of estimates from each study, the inverse of the standard deviation of the mean VSL estimates, and a com- bination of the two—and found that results were reasonably robust to the weights used. More recently, Masterman and Viscusi [32] performed a meta-analysis of global stated preference VSL estimates, using 1,148 estimates drawn from 85 studies. Using least-squares with inverse variance weights and article fixed effects, the authors found large and statistically significant publication biases with bias-adjusted VSLs never larger than $1 million.

To our knowledge, the only existing VSL meta-analysis that combined hedonic wage and stated preference estimates is the study by Kochi et al. [33]. The authors used an empirical Bayes estimation approach in a two-stage pooling model to examine 197 estimates selected from 40 studies published in the U.S. and other high-income countries. In a first stage, the authors created subsets of estimates by the same author or groups of authors and calculated the mean value for the subset if it passed a statistical test for homogeneity. In a second stage, the authors combined estimates from the 60 homogeneous subsets accounting for across- group variability using the Q-statistics for each group. A bootstrap approach was used to compare the distributions of VSL by study type. The authors found that the mean VSL from hedonic wage studies was roughly three times larger than that from stated preference studies.

Summary statistics from the VSL meta-analyses covered in this section appear in Table 1. As a high-level synthesis of the results from prior meta-analyses, we note that the average low, midpoint, and high ends of the ranges reported in the final column of Table 1 are $5.2, $7.4, and $9.5 million. In the Discussion section below, we will compare and contrast our estimation methods and results to some of the studies reviewed here.

**Table 1. Summary statistics from 13 previous VSL meta-analyses.** *I* **is the number of groups,** *N* **is the number of observations, HW and SP indicate whether hedonic wage or stated preference estimates were included, and VSL range spans the smallest to largest estimates reported in the original article (all converted to 2020 US$).**

| Authors (year) | I | N | HW | SP | VSL range |
|---|---|---|---|---|---|
| Mrozek and Taylor (2002) | 33 | 203 | 1 | 0 | 2.4–4.0 |
| Viscusi and Aldy (2003) | 49 | 49 | 1 | 0 | 7.5–9.3 |
| Kochi et al. (2006) | 40 | 197 | 1 | 1 | 4.2–14.4 |
| Bellavance et al. (2009) | 37 | 39 | 1 | 0 | 7.5–12.6 |
| Lindhjem et al. (2011) | 95 | 856 | 0 | 1 | 2.0–9.8 |
| Dekker et al. (2011) | 26 | 77 | 0 | 1 | 3.3–10.7 |
| Doucouliagos et al. (2012) | 37 | 39 | 1 | 0 | 1.3–2.8 |
| Nelson (2015) | 28 | 28 | 1 | 0 | 6.0–13.9 |
| Viscusi (2015) | 17 | 550 | 1 | 0 | 10.6–12.2 |
| Masterman and Viscusi (2017) | 68 | 1025 | 1 | 0 | 9.5–11.5 |
| Viscusi (2018) | 68 | 1025 | 1 | 0 | 8.8–12.4 |
| Masterman and Viscusi (2020) | 85 | 1148 | 0 | 1 | 0.2–1.0 |

With so many VSL meta-analyses now available in the published literature, Banzhaf [34] observed that "...the old problem of selecting a single best study has just been pushed back to the problem of selecting a single best meta-analysis." To consolidate this literature, Banzhaf synthesized 11 meta-estimates of the VSL from 6 prior meta-analyses: one estimate each from USEPA [35], Viscusi and Aldy [22], Robinson and Hammitt [21], two estimates from Mrozek and Taylor [8], two estimates from Kochi et al. [33], and four estimates from Viscusi [30]. In his alternative model, which includes all source studies, Banzhaf gave each study equal weight. He then produced a mixture distribution by taking repeated random draws from the distributions defined by the means and standard errors of the constituent meta-estimates, with the pre-specified weights applied to each estimate. The resulting VSL mixture distribution has a mean of $7.6 million and 90% confidence interval from $2.0 to $13.1 million. We note that Banzhaf's consolidated central estimate is very close to the average of the midpoints in Table 1, and Banzhaf's range safely encompasses the range of average low and high estimates in Table 1.

Our summary of previous meta-analyses above and the quantitative synthesis by Banzhaf point to a similar range of central estimates for the VSL. These preliminaries provide the context for our main goal in the present study, which is to describe and illustrate the use of a multilevel random-effects estimator that is more general and—at least under some circumstances, elaborated below—more precise than those used in many previous VSL meta-analyses.

To set the stage for the technical details of our proposed estimation approach in the following section, we conclude this section by restating the aims and potential benefits of our study. We have developed a new method for synthesizing quantitative estimates from the scientific literature, designed to handle scenarios where studies contribute varying numbers of estimates for the same policy-relevant parameter—a common occurrence in real-world research. By carefully distinguishing between different sources of uncertainty in the data, this method helps to ensure that the final weighted mean estimate is as accurate and precise as possible. Why is this important? For those who rely on meta-analyses to inform or evaluate public policies—such as researchers, policymakers, or students—our method offers a more reliable way to draw meaningful insights from diverse studies. In areas like public health, safety regulations, and environmental policy, decision-makers often use these summaries to assess costs and benefits. More precise and reliable meta-analysis estimators enable smarter, evidence-based decisions, whether determining air quality standards, improving road safety measures, or addressing challenges in other critical domains.

## 2 Methods

Our two-stage random-effects (2SRE) estimation approach is designed to maximize precision using meta-data that take the form of balanced or unbalanced panels, which means it can accommodate primary studies that contribute any number of observations. It also accounts for sampling and non-sampling sources of error both within and across studies. Our approach is related to other multilevel meta-analysis methods, including the hierarchical dependence model described by Hedges et al. [36] and Tipton [37] and the three-level meta-analysis approach described by Konstantopoulos [38]. However, in contrast to the studies by Hedges et al. and Tipton, which focus on robust variance estimation for multilevel models given any weighting scheme, our approach uses the metadata to estimate efficient weights. And in contrast to the study by Konstantopolous, which uses a maximum likelihood approach to estimate a single within-study residual error variance, our approach uses the method of moments to estimate non-sampling error variances for each study without assuming a parametric form for the error distributions. Our methodological contributions include: (1) tailoring

a multilevel random-effects estimator to features that we expect to characterize many VSL meta-datasets, (2) conducting a series of Monte Carlo simulation experiments to examine the performance of the estimator in comparison to several other commonly used meta-analysis estimators in our data environment, and (3) applying the estimator to a preliminary meta-dataset of VSL estimates from revealed and stated preference studies conducted in the United States between 1999 and 2019.

## 2.1 A two-stage random-effects meta-analysis estimator

In this sub-section we describe the 2SRE estimator that we propose to use for synthesizing published VSL estimates. To begin, we decompose each observation into the sum of the true effect size and three error components,

$$y_{ij} = Y + \eta_i + \mu_{ij} + \varepsilon_{ij}, \tag{1}$$

where $y_{ij}$ is observed VSL estimate $j$ from group $i$, $Y$ is the average VSL among the U.S. adult general population (our target of estimation), $\eta_i$ is a group-level non-sampling error, $\mu_{ij}$ is an observation-level non-sampling error, and $\varepsilon_{ij}$ is an observation-level sampling error.

Note that we use "non-sampling errors" to refer to what is often called "heterogeneity" in the meta-analysis literature. For example, Hedges et al. [39] discussed this distinction as follows: "Sampling standard error measures the sampling variation of the estimated effect size but does not reflect non-sampling variations which would occur if the study had used a different population of students or different teachers...," and "The variation among studies is, of course, due in part to random sampling fluctuations as reflected in the sampling standard errors. However, in some cases differences between individual studies exceed several standard errors, presumably reflecting differences in the characteristics of those studies... To study this 'non-sampling' variation we use heterogeneity analysis."

To develop the meta-dataset, the EPA selected primary estimates from published studies based on data samples and model specifications originally designed to identify the average VSL among the entire U.S. adult general population (our target of estimation) or a large subset of the general population—e.g., working adults between the ages of 18 and 65, as in many hedonic wage studies. The estimands in primary studies with non-representative samples will differ from our estimand by an amount that depends on the degree to which their samples are not representative along the relevant dimensions and the association between those sample characteristics and people's marginal willingness to pay for mortality risk reductions. In such cases, the primary estimates would be a biased estimate of our estimand even if it were an unbiased estimate of the average VSL among the subset of the population from which the original sample was drawn. All deviations in the primary VSL estimates stemming from differences in the sampling frames, estimation approaches, forms of estimating equations, selection of exogenous control variables, handling of outliers, and any other idiosyncratic data cleaning and modeling choices among the primary studies—i.e., all sources of variability in the primary VSL estimates that do not arise from sampling variation per se—are subsumed in our composite non-sampling error terms, $\eta_i + \mu_{ij}$.

Next, we decompose the composite errors such that $\eta_i$ varies between but not within groups, while $\mu_{ij}$ and $\varepsilon_{ij}$ can vary both between and within groups. The standard errors reported for each observation represent the sampling variability of the published estimates conditional on the designs of the original studies. We assume the variances of the sampling error components, $\sigma_{\varepsilon,i}^2$, are equal to the squared standard errors of the VSL estimates as

reported in the original studies, $se_{ij}^2$. The variances of the between- and within-group non-sampling error components, $\sigma_\eta^2$ and $\sigma_{\mu,i}^2$, are unknown and will be estimated from the data.

For our meta-analysis estimator to be unbiased, all error components must have means of zero. This is a common assumption but its plausibility will depend in part on the selection criteria used to draw primary estimates from the published literature. In particular, at least two constituent assumptions must hold to make $\mathbb{E}[\eta_i]=0$ and $\mathbb{E}[\mu_{i,j}]=0$: (1) the non-sampling errors stemming from non-representative sampling frames and differences in study designs are idiosyncratic, and so just as likely to lead to positive as negative biases with respect to our estimand, and (2) publication bias is negligible, and so the estimates that appear in the published literature are not selected on their magnitudes. We will maintain the first assumption throughout, but we will demonstrate how to test the second assumption in a side-analysis using two conventional publication bias estimators. (While publication bias is an important issue, it is not the primary focus of our methodological contribution in this study; as such, our discussion of it will remain limited.)

Conditional on the zero-mean-errors assumption, any convex combination of the observations will provide an unbiased estimate of the average VSL. Our aim is to find the set of weights that gives an unbiased estimate with the lowest possible variance. The estimator can be written as a weighted average,

$$\hat{Y} = \sum_{i=1}^{I} \sum_{j=1}^{J_i} w_{ij} y_{ij}, \tag{2}$$

where $\sum_{i=1}^{I} \sum_{j=1}^{J_i} w_{ij} = 1$. We derived formulas for the weights as follows. First, we found conditional observation-level weights, $g_{ij}$, to calculate group-level estimates $\hat{Y}_i = \sum_{j=1}^{J_i} g_{ij} y_{ij}$, where $\sum_{j=1}^{J_i} g_{ij} = 1$. Second, we found group-level weights to compute the overall estimate of the true effect size, $\hat{Y} = \sum_{i=1}^{I} h_i \hat{Y}_i$, where $\sum_{i=1}^{I} h_i = 1$. Third, we calculated the unconditional observation-level weights as $w_{ij} = h_i g_{ij}$. We constrained the weights to sum to 1 at each level, which ensures that the group-level estimates are unbiased and that the overall estimate is unbiased.

We derived the $g_{ij}$'s to minimize the variance of the group-level estimates, which requires estimates of $\sigma_{\mu,i}$ for each group, and we derived the $h_i$'s to minimize the variance of the overall estimate, which depends on the conditional variances of the group-level estimates and requires an estimate of $\sigma_\eta$. We used a method-of-moments approach to derive estimators for $\sigma_{\mu,i}$ and $\sigma_\eta$, so no assumptions about the shapes of the error distributions were required.

Some groups in a meta-dataset may have only a single observation, which means $\sigma_{\mu,i}$ cannot be estimated for those groups. To proxy the within-group non-sampling error variance for singleton groups, we use the average of the $\hat{\sigma}_{\mu,i}$'s for the non-singleton groups. The alternative of assuming $\sigma_{\mu,i} = 0$ for singleton groups would have the unintended effect of penalizing primary studies that reported more than one estimate. By assigning the mean non-sampling error variance to the singleton groups, non-singleton groups with observations that have lower than average non-sampling error variances will receive more weight than the singleton groups, and those with higher than average non-sampling error variances will receive less weight, all else equal. This gives more leverage to studies whose estimates are more robust to variations in functional form assumptions and other sensitivity tests designed to examine uncertainties unrelated to sampling variability.

The estimator also allows for correlation among sampling errors, $\rho$, but does not estimate this value. The analyst must specify $\rho$ and can examine the influence of this assumption through sensitivity analysis. We investigated the effect of mis-specifying this correlation in our Monte Carlo experiments described below.

The foregoing description of the estimation approach has focused on the calculation of precision weights for the observations in a meta-analysis context, with no moderator variables included. For use in meta-regression models, which include one or more moderator variables intended to help explain some of the systematic heterogeneity among the quantities estimated in each primary study, the same approach to calculating the optimal precision weights applies except the true effect size, $Y$, is replaced with $f(x_{ij}, \beta)$—e.g., $x_{ij}\beta$ in a linear meta-regression model—in Eq (1) above. All equations necessary to compute the 2SRE estimator are shown in Table 2, and a full derivation is provided in Sect. S1 of the Supporting information.

In our illustrative application, we used iterated weighted least squares to estimate linear meta-regression models. This involves initializing $\hat{\beta}$ by regressing $y$ on $x$ with no weighting (ordinary least squares) or precision weights based on the reported standard errors only (a fixed-effect size meta-regression model). Then $\hat{\beta}$ is used to estimate the error component variances, and the estimated error component variances are used to recalculate $\hat{\beta}$ using weighted least squares. The process is repeated until the estimates converge to stable values. If $x$ only includes a constant, then the estimator collapses to the simple meta-analysis model described above with no moderator variables, in which case no iteration is required. The iterated least squares estimation procedure steps are shown as a flow diagram in Fig 1.

## 2.2 Performance comparison using Monte Carlo experiments

To examine the performance of the 2SRE estimator, we conducted a series of Monte Carlo simulation experiments using constructed data. For each experiment, we specified the true VSL, $Y$, the number of groups, $I$, the number of observations for each group, $J_i$, the error component variances, $\sigma_\eta^2$ and $\sigma_{\mu,i}^2$, and the within-group sampling error correlation, $\rho$. For 16 combinations of the experimental design parameters, we applied several alternative meta-analysis estimators, including the 2SRE estimator, to each of 2,000 simulated meta-datasets. The estimators we compared are listed and described in Table 3.

The first two estimators, the simple mean and group means, make no use of the reported standard errors for each observation nor do they attempt to estimate any unobserved error components for precision weighting. The next three estimators—metafor, robumeta, and MAd—are commonly used meta-analysis packages developed for R. The final estimator is the the two-stage random-effects estimator developed in this study. We applied three versions of the 2SRE estimator. The first version (2SRE-true) uses the true error component variances to compute precision weights. This is impossible using real data, but is useful here to provide a theoretical lower bound estimate of the standard errors for all feasible estimators that can take the form of an unbiased weighted mean as in Eq (2). The second version (2SRE-free) allows for heterogeneous within-group non-sampling error variances. This version is the most general and should be the most efficient feasible estimator with sufficiently many groups and observations per group. The third version (2SRE-equal) is constrained by imposing a common within-group non-sampling error variance. This version may outperform the second version of the 2SRE estimator if the number of observations per group is small.

The precision of each estimator is indicated by the standard deviation of the resulting VSL weighted mean estimates among all 2,000 Monte Carlo trials. For comparison to our simulation-based estimates of standard errors, we also calculated robust standard errors following Hedges et al. [36].

## 2.3 Detecting and adjusting for publication bias

A common concern in meta-analyses is the possibility of publication bias [45]. Though our main focus in this study is on the statistical efficiency of alternative meta-analysis estimators

**Table 2. All equations necessary to compute the two-stage random-effects (2SRE) estimator listed in a feasible sequence. A complete derivation is provided in the Supplemental Information.**

$$\hat{\sigma}^2_{\mu,i} = \frac{1}{J_i - 1} \sum_{j=1}^{J_i} \left( y_{i,j} - \frac{1}{J_i} \sum_{j=1}^{J_i} y_{i,j} \right)$$

$$-\frac{1}{J_i} \sum_{j=1}^{J_i} \left( se^2_{i,j} - \frac{1}{J_i} \sum_{k \neq j}^{J_i} \rho_i se_{ij} se_{i,k} \right)$$

2.1

$$\hat{A}_{i,j} = \left( \left[ \hat{\sigma}^2_{\mu,i} + (1 - \rho_i) se^2_{i,j} \right] \sum_{j=1}^{J_i} \frac{1}{\hat{\sigma}^2_{\mu,i} + (1 - \rho_i) se^2_{i,k}} \right)^{-1}$$

$$\hat{\mathbf{A}}_i = \begin{bmatrix} \hat{A}_{i,1} & \hat{A}_{i,2} & \cdots & \hat{A}_{i,J_i} \end{bmatrix}'$$

2.2

$$\hat{B}_{i,j,k} = \rho_i \left( \frac{\sum_{\ell=1}^{J_i} \frac{se_{i,\ell}}{\hat{\sigma}^2_{\mu,i} + (1-\rho_i) se^2_{i,\ell}}}{\sum_{\ell=1}^{J_i} \frac{\hat{\sigma}^2_{\mu,i} + (1-\rho_i) se^2_{i,j}}{\hat{\sigma}^2_{\mu,i} + (1-\rho_i) se^2_{i,\ell}}} - \frac{se_{i,j}}{\hat{\sigma}^2_{\mu,i} + (1 - \rho_i) se^2_{i,j}} \right) se_{i,k}$$

$$\hat{\mathbf{B}}_i = \begin{bmatrix} \hat{B}_{i,1,1} & \hat{B}_{i,1,2} & \cdots & \hat{B}_{i,1,J_i} \\ \hat{B}_{i,2,1} & \hat{B}_{i,2,2} & \cdots & \hat{B}_{i,2,J_i} \\ \vdots & \vdots & \ddots & \vdots \\ \hat{B}_{i,J_i,1} & \hat{B}_{i,J_i,2} & \cdots & \hat{B}_{i,J_i,J,i} \end{bmatrix}$$

2.3

$$\hat{\mathbf{g}}_i \equiv \begin{bmatrix} \hat{g}_{i,1} & \hat{g}_{i,2} & \cdots \hat{g}_{i,J_i} \end{bmatrix}' = \left( \mathbf{I}_i - \hat{\mathbf{B}}_i \right)^{-1} \hat{\mathbf{A}}_i$$

2.4

$$\hat{Y}_i = \sum_{j=1}^{J_i} \hat{g}_{i,j} y_{i,j}$$

2.5

$$\hat{\sigma}^2_\eta = \frac{1}{I} \sum_{i=1}^{I} \left( \hat{Y}_i - \frac{1}{I} \sum_{i=1}^{I} \hat{Y}_i \right)^2$$

$$-\frac{1}{I} \sum_{i=1}^{I} \sum_{j=1}^{J_i} \left[ \hat{g}^2_{i,j} \left( \hat{\sigma}^2_{\mu,i} + se^2_{i,j} \right) \rho_i \sum_{k \neq j}^{J_i} \hat{g}_{i,j} \hat{g}_{i,k} se_{i,j} se_{i,k} \right]$$

2.6

$$\hat{v}_i = \hat{\sigma}^2_\eta + \sum_{j=1}^{J_i} \left[ \hat{g}^2_{i,j} \left( \hat{\sigma}^2_{\mu,i} + se^2_{i,j} \right) \rho_i \sum_{k \neq j}^{J_i} \hat{g}_{i,j} \hat{g}_{i,k} se_{i,j} se_{i,k} \right]$$

2.7

$$\hat{h}_i = \frac{\hat{v}_i^{-1}}{\sum_{q=1}^{I} \hat{v}_q^{-1}}$$

2.8

$$\hat{w}_{i,j} = \hat{g}_{i,j} \hat{h}_i$$

2.9

$$\hat{Y} = \sum_{i=1}^{I} \hat{h}_i \hat{Y}_i = \sum_{i=1}^{I} \sum_{j=1}^{J_i} \hat{w}_{i,j} y_{i,j}$$

2.10

Meta-regression: When conducting a meta-analysis with moderator variables, $x_{i,j}$, the first term on the right-hand side of Eq (2.1), should be replaced with $\frac{1}{J_i-1} \sum_{j=1}^{J_i} \left[ \left( y_{i,j} - \sum_{j=1}^{J_i} y_{i,j} \right)^2 - \left( x_{i,j} \hat{\beta}_j - \sum_{j=1}^{J_i} x_{i,j} \hat{\beta}_j \right)^2 \right]$. The resulting $\hat{w}_{i,j}$'s can be used as weights in a weighted-least-squares estimation of $\beta$. Start at $\hat{\beta} = 0$, iterate the entire sequence of computations updating $\hat{\beta}$ each iteration, and repeat until convergence.

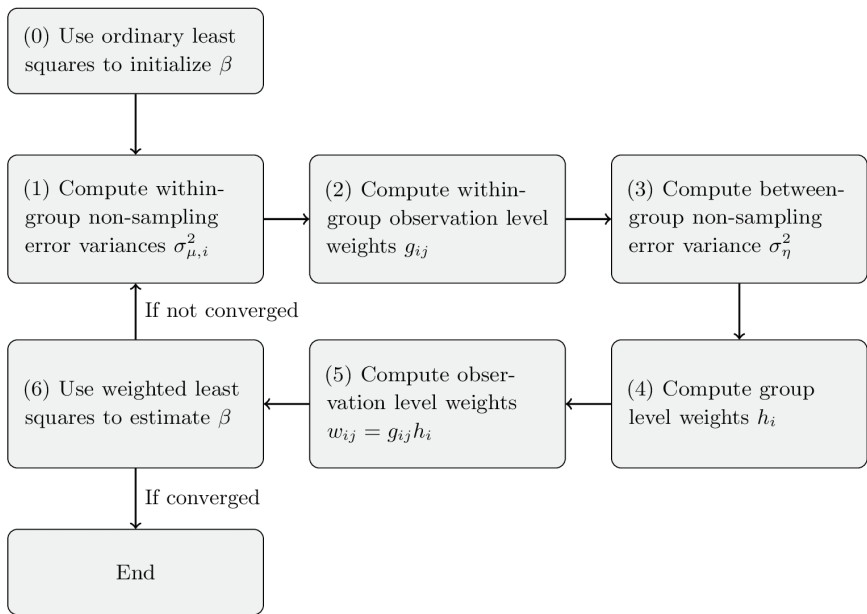

**Fig 1. Flow diagram of iterated least squares.** Steps required to compute the two-stage random-effects meta-regression estimator, with moderator variables *x*, using iterated least squares. Detailed equations are presented in Table 2. For meta-analysis, with no moderator variables, step (6) involves computing the overall weighted mean, $\hat{Y}$, rather than estimating the moderator coefficients, $\beta$, and no iteration is required.

**Table 3. Alternative meta-analysis estimators compared in Monte Carlo simulation experiments.**

| Estimator | Description |
|---|---|
| Simple mean | Unweighted mean of all observations |
| Group means | Unweighted mean of group means |
| Metafor | R package for estimating meta-regression models [40,41]. Uses restricted maximum likelihood to estimate $\sigma_\eta$ assuming normally distributed errors. Accounts for $\hat{\rho}$ in the `rma.mv` function via user-specified variance-covariance matrix of sampling errors. Does not estimate within-group non-sampling error variances, $\sigma_{\mu,i}$. |
| Robumeta | R package for meta-regression with robust (Huber-White) standard errors for non-independent observations [42]. We estimated two versions: (1) The correlated effects model (CORR) computes approximate inverse variance weights assuming constant error variances within groups and a user-specified common within-group correlation, $\rho$. (2) The hierarchical effects model (HIER) corresponds to our 2SRE estimator with $\sigma_{\mu,i} = \sigma_\mu \forall i$ and $\rho = 0$, but appears to use different estimators for the error component variances. |
| MAd | R package that provides a wrapper for metafor [43,44]. Includes a procedure to aggregate dependent observations with a user-specified within-group correlation. |
| 2SRE | Two-stage random-effects meta-analysis estimator, as described in the main text and Supporting information. We estimated three versions: (1) "2SRE-true" using inverse variance weights computed with the true error component variances, (2) "2SRE-free" using estimated error component variances with $\sigma_{\mu,i}$ estimated freely for each group, and (3) "2SRE-equal" using estimated error component variances with $\sigma_{\mu,i}$ constrained to be equal for all groups. |

when applied to VSL meta-datasets, we also used two conventional methods to address publication bias: the trim-and-fill and PET-PEESE estimators.

The trim-and-fill estimator [46] is a non-parametric method based on the observation that a plot of precision estimates ($1/se^2$) versus corresponding effect size estimates—often called

a "funnel plot"—should be vertically symmetric. If all estimates are equally likely to be published, then the funnel plot should be wide at the bottom (low precision studies) and narrow at the top (high precision studies) with roughly the same number of estimates on the left and right sides of their center of mass. On the other hand, if estimates with low $t$-statistics are less likely to be published, then the funnel plot will have a conspicuously lower density of estimates in the bottom-left region of the plot (assuming positive effect size estimates). The trim-and-fill estimator works by iteratively "trimming" estimates on the far right side of the plot until the trimmed funnel is no longer asymmetric, then re-calculating the mean of the remaining estimates, then "filling" the trimmed and missing estimates on both sides of the plot around the corrected mean to compute the variance of the estimator.

The PET-PEESE estimator [47] uses a two-stage regression approach to detect and correct for publication bias. The first stage (the PET or "precision effect test") involves regressing the effect size estimates on a constant and the standard errors. If the coefficient on the standard errors is significantly different from zero, this is taken as evidence of publication bias. In these cases, a second stage (the PEESE or "precision-effect estimate with SE") is applied, which involves regressing the effect size estimates on a constant and the squared standard errors. The estimated constant in this regression is taken as a corrected mean effect size. Intuitively, the $se^2$ term controls for the influence of study precision on the reported effect size estimates, and the estimated constant extrapolates the relationship to an (hypothetical) infinitely precise study with $se^2 = 0$. Newer methods for detecting and addressing publication bias have been proposed [48,50]. Integrating our meta-analysis estimator with these approaches is a more complex task that we do not attempt here, so we flag this as an important direction for future research.

## 2.4 Application to a previously published VSL meta-dataset

To demonstrate the 2SRE estimation approach using realistic data, we applied it to a previously published meta-dataset assembled by the U.S. Environmental Protection Agency for a review of proposed meta-analysis methods by the Agency's Science Advisory Board [16]. The EPA dataset included studies published up to 2013. To update it, we added one additional study published since then that met the same selection criteria [49]. While a more comprehensive VSL meta-dataset could be constructed by relaxing certain selection criteria, we chose to use the EPA meta-dataset for two key reasons. First, it is pre-existing, freely accessible, and thoroughly documented in an online government report, which includes details on the search strategy, screening criteria, and a PRISMA diagram. Second, our primary aim in this paper is methodological rather than empirical. We are not attempting to generate a synthesized VSL estimate for policy use by the EPA or other agencies; instead, our focus is on evaluating the performance of the proposed estimation approach. For these reasons, using a previously published and widely available meta-dataset, albeit not exhaustive, is most suitable for our purposes.

The dataset contains VSL estimates (hereafter "observations") from both revealed preference and stated preference studies. Multiple observations were drawn from studies meeting the screening criteria. Where available, these include both mean and median VSL estimates and their respective standard errors. The dataset is composed of 46 observations from 9 hedonic wage studies, 25 observations from 1 quasi-experimental study, and 42 observations from 9 stated preference studies. Detailed information about the dataset, including the full list of studies through 2013 and the screening criteria, is provided by USEPA [16].

The EPA Science Advisory Board made a number of recommendations for altering both the dataset and methods proposed in the 2015 EPA report [51]. An important motivation

for the present study was the board's recommendations to refine and improve the estimation approach. The purpose of our analysis of the preliminary meta-dataset was to demonstrate the proposed estimation approach using realistic data. Given the preliminary nature of the dataset, the results presented here should be viewed as illustrative and do not represent an official summary measure of the VSL for use in benefit-cost analysis.

## 3 Results

### 3.1 Monte Carlo experiments

We compared the candidate estimators under four combinations of true and assumed correlations among non-sampling errors within studies, $\rho$ and $\hat{\rho}$. In all four combinations, we examined 16 unique combinations of the number of groups, $I$ (20 or 60), the minimum and maximum number of observations in each group, $J$ (drawn randomly from the range 1–5 or 1–15), the group-level (between groups) non-sampling error variability, $\sigma_\eta$ (1.0 or 3.0), and the observation-level (within groups) non-sampling error variability, $\sigma_\mu$ (drawn randomly from the range 0.5–1.0 or 0.5–3.0). In all cases the true VSL was 10 and the sampling error variability, $se$, was drawn from the range 0.5–5.0.

Vertical box plots summarizing the relative performance of all tested estimators are shown in Fig 2. The interior line and the top and bottom edges of each box correspond to the means and ranges of the standard errors of each estimator normalized by their theoretical minimum possible standard errors (based on the 2SRE-true estimator, which uses the true error component variances to compute standard errors). For example, a box with a center line at height 0.2 indicates that the average ratio of the standard error to the minimum possible standard error among the 16 combinations of design settings was 1.2. The top and bottom edges of each box indicate the minimum and maximum normalized standard errors for each estimator across all 16 combinations examined in our Monte Carlo experiments. These charts show that the 2SRE-equal estimator performs at least as well as the others on average in all four $(\rho, \hat{\rho})$ combinations. The charts also show that the 2SRE-free estimator performs poorly relative to the constrained version, especially when $\rho=0$. The simple mean and group mean estimators show their best performance when $\rho=0$. The efficiency advantages of the more sophisticated meta-analysis estimators are more clearly evident when $\rho>0$, a condition we expect to hold in most realistic meta-datasets that include multiple estimates from the same study or the same underlying primary datasets.

In all cases, the data were constructed with $\sigma_{\mu,i}$ heterogeneous across groups, so the constrained 2SRE-equal estimator, with $\hat{\sigma}_{\mu,i} = \hat{\sigma}_\mu$ for all $i$, imposes a binding restriction on the estimating equation. This restriction will not bias the estimator but will make it less efficient than the unconstrained 2SRE-free estimator in sufficiently large samples, or more efficient in sufficiently small samples, where the large-versus-small sample size threshold will depend on all parameters of the data generating process. We attempted to vary the experimental design settings to cover ranges that are typical for VSL meta-analyses, so the Monte Carlo comparisons of the estimators are meant to be informative for realistic VSL meta-analysis applications.

Detailed Results from our Monte Carlo experiments including all 16 cases under all four $(\rho, \hat{\rho})$ combinations are shown in Tables S2.1–S2.4 in the Supporting information. The supplemental tables show that the estimated standard errors are less than 1.0 for nearly all estimators under nearly all experimental design settings. This is relatively high precision considering that the true VSL was set at 10 for these numerical experiments. Therefore, using a VSL meta-dataset with characteristics within the range of sample sizes and error component variances

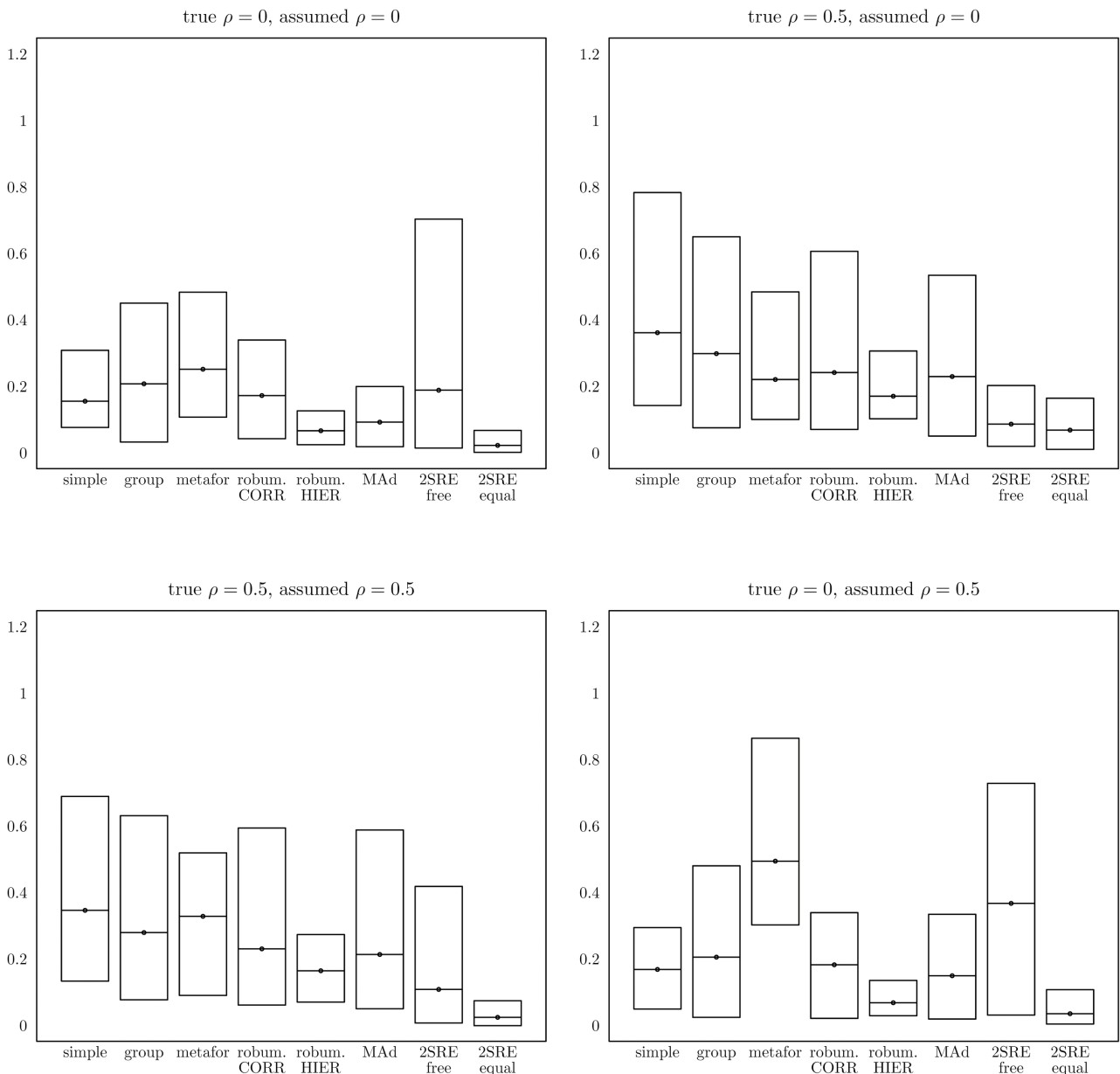

**Fig 2. Estimator performance comparisons.** The center, bottom, and top horizontal lines of each box are the average, minimum, and maximum relative precision measures for each estimator across the 16 experimental design settings tested. The relative precision was computed as the difference between the standard error of the estimator and the minimum theoretical standard error divided by the minimum theoretical standard error, i.e., $\left(\hat{se} - se_{min}\right) / se_{min}$. Therefore, lower box heights indicate better performance.

considered here, a variety of reasonable meta-analysis estimators should produce a 95% confidence interval with a half-width less than 20% of the central estimate itself. Nevertheless, the box plots in Fig 2 and the tables in Supporting information Sect. S2 clearly show systematic differences in performance among the competing estimators. In particular, the 2SRE-equal

estimator is more precise than most other estimators under most experimental design configurations, the robumeta hierarchical estimator also performs well, and the 2SRE-free estimator performs poorly especially when $\rho = 0$.

## 3.2 Demonstration using realistic data

*Meta-analysis results*

A variety of meta-analysis estimates using several subsets of the preliminary EPA metadata are shown in Table 4. Results for seven estimators are presented: simple mean, group means, 2SRE-free, 2SRE-equal, and three modified versions of the 2SRE estimator with corrections for publication bias using the trim-and-fill (T&F) and the PET-PEESE (P-P) methods. "mm" indicates that both mean and median VSL observations were included; "m" indicates that only mean VSL observations were included. Each estimator was applied to data only from revealed preference (RP) studies, only from stated preference (SP) studies, and from both RP and SP studies (pooled). The final column shows the simple average of the independent RP and SP estimates (balanced), which places equal weight on the two types of primary estimation methods regardless of the number of studies and observations of each type. Numbers in parentheses are bootstrapped standard errors, and numbers in brackets are root mean squared errors (RMSE's) where the bias was estimated as the difference between the "mm" and "m" estimates. The "m" estimates are assumed to be unbiased, in which cases the root mean squared errors are equal to the standard errors.

When comparing primary estimation approaches, the RP estimates are larger than the SP estimates in 12 of 14 cases, but the differences are smaller when using only mean VSL observations from the SP studies. The pooled and balanced estimates are very close to each other for all estimators that do not involve publication bias corrections. The largest difference between the pooled and balanced estimates is produced by the 2SRE-free T&F estimator using only mean VSL observations, for which the balanced estimate is nearly $1.5 million larger than the pooled estimate.

All primary studies using a revealed preference approach reported only mean VSL observations, so the "mm" and "m" entries are the same for each estimator in the RP column. Primary studies using stated preference approaches reported mean or median or both types of VSL observations, so the "mm" and "m" entries are different for each estimator in the SP column. In all cases, median observations were lower than mean observations, so the "mm" estimates are lower than the "m" estimates. Our target of estimation was the mean VSL among the adult U.S. population, so the "m" estimates are assumed to be unbiased. Pooling mean and median observations biases our estimates down, but also reduces the variance of the estimates by virtue of the larger sample sizes. Treating the "m" estimates as unbiased and computing the root mean squared error (RMSE) as the square root of the sum of the squared standard errors plus the square of the difference between the focal estimate and the "m" estimate, we find that in 4 of 6 cases the RMSE's of the "mm" SP estimates are lower than those of the "m" SP estimates. This suggests that on the mean squared error criterion, pooling mean and median observations can be advantageous in this setting.

Publication bias corrections have variable effects on the meta-analysis estimates. The trim-and-fill (T&F) correction reduces the 2SRE RP estimates by $0.36 and $0.57 million, and it reduces most of the 2SRE SP estimates by $2 million or more, the only exception being the 2SRE-free "m" estimate which increases slightly. The PET-PEESE (P-P) HW estimate is $1.4 and $0.83 million lower than the uncorrected 2SRE estimates, and the P-P SP estimates are $0.56 lower and $0.92 million higher than the corresponding uncorrected 2SRE-equal "mm" and "m" estimates.

**Table 4. Meta-analysis estimates of the average VSL among US adults [2020$US] based on the preliminary meta-dataset compiled by the [16].** Four estimates with the lowest RMSE's are highlighted in bold font. "mm" rows indicate datasets including both mean and median primary estimates. "m" rows indicate datasets including only mean primary estimates. The "RP" column indicates datasets including only revealed preference estimates, the "SP" column indicates datasets including only stated preference estimates, the "pooled" column includes both RP and SP estimates, and the "balanced" column includes both types of estimates with equal weight placed on each.

| Estimator | m(m) | RP | | SP | | | pooled | | | balanced | | |
|---|---|---|---|---|---|---|---|---|---|---|---|---|
| simple mean | mm | 10.73 | (1.78) | 7.65 | (1.16) | [1.58] | 9.59 | (1.30) | [1.42] | 9.19 | (0.74) | [0.91] |
| | m | 10.73 | (1.78) | 8.72 | (2.12) | | 10.16 | (1.48) | | 9.73 | (0.98) | |
| group means | mm | 10.70 | (1.24) | 8.14 | (1.12) | [1.56] | 9.49 | (0.90) | [1.11] | **9.42** | (0.59) | [0.80] |
| | m | 10.70 | (1.24) | 9.22 | (1.85) | | 10.14 | (1.03) | | 9.96 | (0.77) | |
| 2SRE–free | mm | 8.41 | (0.76) | 6.64 | (0.69) | [1.50] | **7.61** | (0.49) | [0.72] | **7.52** | (0.36) | [0.76] |
| | m | 8.41 | (0.76) | 7.97 | (1.64) | | 8.15 | (0.72) | | 8.19 | (0.60) | |
| –equal | mm | 8.50 | (0.76) | 7.20 | (1.28) | [1.94] | 7.91 | (0.84) | [1.17] | 7.85 | (0.51) | [0.89] |
| | m | 8.50 | (0.76) | 8.66 | (1.82) | | 8.72 | (0.97) | | 8.58 | (0.65) | |
| –free T& F | mm | 7.76 | (1.06) | 4.65 | (1.08) | [3.62] | 5.53 | (1.22) | [1.25] | 6.21 | (0.54) | [1.81] |
| | m | 7.76 | (1.06) | 8.11 | (2.57) | | 5.79 | (1.91) | | 7.93 | (0.91) | |
| –equal T& F | mm | 7.76 | (1.06) | 4.35 | (1.72) | [1.81] | 4.83 | (1.47) | [1.52] | **6.06** | (0.70) | [0.75] |
| | m | 7.76 | (1.06) | 4.90 | (2.22) | | 5.22 | (1.76) | | 6.33 | (0.82) | |
| –P-P | mm | 7.31 | (0.99) | 6.28 | (3.11) | [4.28] | 6.35 | (3.03) | [4.15] | 6.79 | (1.03) | [1.79] |
| | m | 7.31 | (0.99) | 9.22 | (3.52) | | 9.19 | (3.34) | | 8.27 | (1.13) | |

A broad-brush summary of the results in Table 4 is that the average of all estimates is $8.14 million, and 38 of 49 estimates (not counting the repeated RP estimates) are between $6 and $10 million, including the four estimates with the lowest RMSE's highlighted in bold font.

*Meta-regression results*

In addition to the meta-analysis results reported in Table 4, we also estimated a variety of meta-regression specifications with control variables for SP observations, median observations, the year of data collection, and the average U.S. income in the year of data collection. We estimated a benchmark model with no control variables plus six specifications including two or more control variables or their interactions. Beginning with Table 5, we show results for the following seven specifications:

- s0. No controls
- s1. SP, median
- s2. SP, median, year
- s3. SP, median, income
- s4. SP, median, year, income
- s5. SP, median, year, SP×year
- s6. SP, median, income, SP×income

Table 6 shows results from seven parallel specifications where each also includes the standard error of the primary VSL observations, *se*, as an additional control variable, which implements the PET stage of the PET-PEESE publication bias estimator. Table 7 shows results for the same specifications where each also includes the squared standard error, $se^2$, which implements the PEESE stage of the PET-PEESE estimator. In Tables 5–7, the 2SRE-equal meta-regression estimation approach was used. Tables S2.5–S2.7 in the Supporting information show the same specifications as the preceding three tables but using the 2SRE-free estimation approach.

In Table 5, the estimate of the constant in specification s0 matches the 2SRE-equal pooled "mm" estimate in Table 4. This occurs because the meta-regression estimator with no control variables is equivalent to the meta-analysis estimator. The standard errors are slightly different because in Table 4 we report bootstrapped standard errors while in Table 5 we report robust standard errors. All estimates of $\sigma_\mu$ and $\sigma_\eta$ in Table 5 are between 2 and 3, which is within the ranges of values used in our Monte Carlo simulation experiments. Coefficient estimates for

**Table 5. EPA data meta-regression results with no correction for publication bias. Seven specifications (s0–s6) of the two-stage random-effects meta-regression model with $\sigma_\mu$ constrained (2SRE-equal) and with income elasticity of VSL *(IEVSL)* for models including income. Numbers in parentheses are robust standard errors.**

|  | s0 | s1 | s2 | s3 | s4 | s5 | s6 |
|---|---|---|---|---|---|---|---|
| constant | 7.907 | 8.894 | 9.928 | 8.802 | 9.813 | 9.621 | 9.544 |
|  | (0.847) | (0.862) | (0.745) | (1.130) | (0.993) | (0.629) | (0.830) |
| SP |  | −0.583 | −3.492 | −0.327 | −3.174 | −3.543 | 0.731 |
|  |  | (2.295) | (1.373) | (2.490) | (1.789) | (1.363) | (2.279) |
| median |  | −2.130 | −1.160 | −2.196 | −1.241 | −0.977 | −2.915 |
|  |  | (2.194) | (1.365) | (2.140) | (1.386) | (1.364) | (2.042) |
| year |  |  | 0.592 |  | 0.593 | 0.417 |  |
|  |  |  | (0.129) |  | (0.130) | (0.039) |  |
| income |  |  |  | −0.098 | −0.122 |  | 0.691 |
|  |  |  |  | (0.579) | (0.521) |  | (0.254) |
| SP×year |  |  |  |  |  | 0.288 |  |
|  |  |  |  |  |  | (0.190) |  |
| SP×income |  |  |  |  |  |  | −1.867 |
|  |  |  |  |  |  |  | (0.771) |
| *se* |  |  |  |  |  |  |  |
| *se²* |  |  |  |  |  |  |  |
| $\sigma_\mu$ | 2.168 | 2.164 | 2.097 | 2.162 | 2.093 | 2.097 | 2.122 |
| $\sigma_\eta$ | 2.733 | 2.735 | 2.755 | 2.736 | 2.756 | 2.755 | 2.748 |
| *IEVSL* |  |  |  | −0.055 | −0.068 |  | 0.386 |
|  |  |  |  | (0.323) | (0.291) |  | (0.142) |
| $R^2$ | 0.560 | 0.600 | 0.749 | 0.600 | 0.749 | 0.757 | 0.635 |
| $R^2_{CV}$ | 0.542 | 0.532 | 0.698 | 0.520 | 0.688 | 0.696 | 0.551 |

the SP dummy variables are always negative, but their magnitudes vary widely across specifications (from –$0.58 to –$3.5 million). The SP coefficient estimate is statistically significant only in specifications s2 and s5. Coefficient estimates for the median dummy variable are always negative and between –$1 and –$3 million, but never statistically significant. The time trend is between $0.4 and $0.6 million per year, and is statistically significant in all three specifications in which it appears. The income coefficient is negative in specifications s3 and s4 but not statistically significant; it is positive and statistically significant in specification s6. The corresponding value of the *IEVSL* at the means of the control variables in specification s6 is 0.386. Based on the $R^2_{CV}$ values reported in the final row of Table 5—which were computed using leave-one-out cross-validation residuals [52]—the best-fitting specification is s2, and the best-fitting specification that excludes the time trend variable is s6. The EPA Science Advisory Board recommended that, in the absence of a clear rationale for giving different weights to estimates from different years, a time trend should not be included in the specification. Instead, they suggested that the influence of the timing of the studies be explored through sensitivity analysis [51].

In Table 6, the *se* coefficient is close to the conventional threshold for statistical significance (somewhat above or below a *t*-statistic of 2) in all specifications. We view this as modest evidence for publication bias according to the PET test.

In Table 7, the PEESE-corrected estimates of the constant—which correspond to RP-based VSL observations at the average of the 'datayear' variable—in all specifications are lower than their counterparts in the benchmark specifications reported in Table 5, where the differences are between $0.6 and $1.0 million. However, the results in Tables 6 and 7 should be viewed in light of the relatively noisy PET-PEESE meta-analysis estimates reported in Table 4 above,

**Table 6. EPA data meta-regression results with the "precision-effect test" (PET) for publication bias. Seven specifications (s0–s6) of the two-stage random-effects meta-regression model with $\sigma_\mu$ constrained (2SRE-equal) and with *IEVSL* for models including income. Numbers in parentheses are robust standard errors.**

|  | s0 | s1 | s2 | s3 | s4 | s5 | s6 |
|---|---|---|---|---|---|---|---|
| constant | 6.373 | 6.544 | 8.237 | 6.407 | 8.093 | 7.621 | 7.355 |
|  | (1.484) | (1.198) | (1.283) | (1.410) | (1.414) | (1.080) | (1.406) |
| SP |  | 0.953 | −2.251 | 1.322 | −1.864 | −2.158 | 1.681 |
|  |  | (2.129) | (1.353) | (2.249) | (1.712) | (1.359) | (2.222) |
| median |  | −2.251 | −1.300 | −2.345 | −1.399 | −1.083 | −2.780 |
|  |  | (2.305) | (1.396) | (2.253) | (1.412) | (1.398) | (2.232) |
| year |  |  | 0.553 |  | 0.553 | 0.318 |  |
|  |  |  | (0.162) |  | (0.163) | (0.066) |  |
| income |  |  |  | −0.139 | −0.148 |  | 0.372 |
|  |  |  |  | (0.494) | (0.471) |  | (0.333) |
| SP×year |  |  |  |  |  | 0.376 |  |
|  |  |  |  |  |  | (0.224) |  |
| SP×income |  |  |  |  |  |  | −1.195 |
|  |  |  |  |  |  |  | (0.818) |
| *se* | 0.968 | 0.950 | 0.653 | 0.952 | 0.655 | 0.737 | 0.763 |
|  | (0.520) | (0.425) | (0.416) | (0.431) | (0.420) | (0.382) | (0.461) |
| $se^2$ |  |  |  |  |  |  |  |
| $\sigma_\mu$ | 2.017 | 2.011 | 1.963 | 2.000 | 1.952 | 1.969 | 1.990 |
| $\sigma_\eta$ | 2.778 | 2.779 | 2.793 | 2.782 | 2.796 | 2.791 | 2.785 |
| *IEVSL* |  |  |  | −0.078 | −0.083 |  | 0.208 |
|  |  |  |  | (0.276) | (0.263) |  | (0.186) |
| $R^2$ | 0.634 | 0.655 | 0.773 | 0.656 | 0.775 | 0.786 | 0.667 |
| $R^2_{CV}$ | 0.596 | 0.577 | 0.704 | 0.569 | 0.695 | 0.710 | 0.574 |

as well as the apparently reduced power of the PET-PEESE estimator in random effects panel data environments reported in some simulation studies [53–55].

## 4 Discussion

We described and demonstrated a two-stage random effects (2SRE) meta-analysis estimation approach that accommodates unbalanced panels with single or multiple observations per group, accounts for sampling and non-sampling sources of error, and allows for correlations among non-sampling and sampling errors within groups. Our estimation approach is related to the three-level meta-analysis approach described by Konstantopoulos[38] and to the robust variance estimation approach described by Hedges et al. [36] and Tipton [37], which is operationalized in the robumeta R package [42]. The primary contributions of the present study include our our estimation of efficient observation-level weights using a method-of-moments estimation approach, extensive simulation experiments, and our application of the estimation approach to a realistic, albeit preliminary, VSL meta-dataset, which together provide a robust indication of the strong performance of the estimator in relevant data environments.

We examined the performance of the estimator on constructed datasets in a series of Monte Carlo simulation experiments designed to bracket the range of data features that we expect to characterize VSL meta-analyses focused on primary studies conducted in the U.S. We found that the estimator performs well in this setting compared to alternatives including three meta-analysis estimators that have been developed into commonly-used R packages. The strong performance of the 2SRE estimator included cases involving within-group correlations among sampling errors that the analyst may not correctly specify. The constrained 2SRE-equal estimator, which assumes a common non-sampling error variance among groups,

**Table 7. EPA data meta-regression results with the "precision-effect estimate with SE" (PEESE) for publication bias. Seven specifications (s0–s6) of the two-stage random-effects meta-regression model with $\sigma_\mu$ constrained (2SRE-equal) and with *IEVSL* for models including income. Numbers in parentheses are robust standard errors.**

|          | s0      | s1      | s2      | s3      | s4      | s5      | s6      |
|----------|---------|---------|---------|---------|---------|---------|---------|
| constant | 7.317   | 7.912   | 9.147   | 7.771   | 9.000   | 8.655   | 8.532   |
|          | (0.933) | (0.822) | (0.825) | (1.083) | (1.016) | (0.675) | (0.866) |
| SP       |         | 0.170   | −2.796  | 0.552   | −2.399  | −2.774  | 1.327   |
|          |         | (2.242) | (1.315) | (2.400) | (1.685) | (1.319) | (2.269) |
| median   |         | −2.031  | −1.136  | −2.127  | −1.236  | −0.898  | −2.748  |
|          |         | (2.249) | (1.381) | (2.194) | (1.402) | (1.372) | (2.122) |
| year     |         |         | 0.563   |         | 0.564   | 0.334   |         |
|          |         |         | (0.140) |         | (0.141) | (0.053) |         |
| income   |         |         |         | −0.144  | −0.151  |         | 0.531   |
|          |         |         |         | (0.552) | (0.506) |         | (0.296) |
| SP×year  |         |         |         |         |         | 0.370   |         |
|          |         |         |         |         |         | (0.203) |         |
| SP×income|         |         |         |         |         |         | −1.579  |
|          |         |         |         |         |         |         | (0.801) |
| *se*     |         |         |         |         |         |         |         |
| *se*$^2$ | 0.122   | 0.107   | 0.105   | 0.107   | 0.080   | 0.090   | 0.094   |
|          | (0.045) | (0.037) | (0.044) | (0.038) | (0.048) | (0.042) | (0.038) |
| $\sigma_\mu$ | 2.045 | 2.063 | 2.046 | 2.052 | 2.039 | 2.042 | 2.060 |
| $\sigma_\eta$ | 2.770 | 2.765 | 2.770 | 2.768 | 2.772 | 2.771 | 2.766 |
| *IEVSL*  |         |         |         | −0.081  | −0.084  |         | 0.297   |
|          |         |         |         | (0.309) | (0.283) |         | (0.165) |
| $R^2$    | 0.624   | 0.645   | 0.770   | 0.646   | 0.771   | 0.783   | 0.667   |
| $R^2_{CV}$ | 0.590 | 0.566   | 0.699   | 0.556   | 0.688   | 0.706   | 0.575   |

outperformed the 2SRE-free variant in all of the simulation experiments we conducted. The latter is the most general version of the estimator, which in principle should perform best in large samples. This suggests that our simulated meta-datasets were too small for its potential performance advantages to emerge. The 2SRE-equal variant of the estimator performed best overall in the four cases we examined. In particular, the 2SRE-equal estimator outperformed all others when $\rho$=0.5 and $\hat\rho$=0.5. We believe $\rho$=0.5 is more realistic than $\rho$=0, so we would recommend $\hat\rho$=0.5 as a default setting and the 2SRE-equal variant as a default model due to its superior performance in the range of data environments examined here. Our simulation results also suggest that robust standard errors are (nearly) unbiased even when the analyst incorrectly specifies the correlation among non-sampling errors within groups.

We applied several variations of the 2SRE estimator to a preliminary meta-dataset of VSL estimates assembled by the U.S. EPA. Variations of the estimation approach, including un-weighted and weighted meta-analyses and meta-regressions with and without adjustments for publication bias, were applied to the full dataset and various subsets of the data and produced central estimates of the VSL between $6 and $12 million.

Since the meta-data we used in this study are preliminary, our resulting estimates should be viewed as demonstrative rather than definitive. Yet, as in our illustration, meta-analyses in many other settings—illustrative or otherwise—also may produce multiple estimates like those that appear in Tables 4–6 and Tables S2.5–S2.7. How might an analyst curate or synthesize such a collection of estimates for use in policy evaluations? This will undoubtedly require a number of judgment calls based on the facts of the case at hand. Again using our application as an illustration, we might begin by considering the pooled and balanced meta-analysis estimates that do not correct for publication bias in Table 4 and the meta-regression estimates in

Tables 5 and 7 because these use all of the available evidence from both revealed and stated preference studies. Among the estimates in Table 4, we would focus on those with the lowest root mean squared errors, which include the group means "mm" balanced case ($\hat{Y}$ = 9.42, RMSE = 0.80), the 2SRE-free "mm" pooled case ($\hat{Y}$ = 7.61, RMSE = 0.72), and the 2SRE–free "mm" balanced case ($\hat{Y}$ = 7.52, RMSE = 0.76). Next, we would consider the corresponding 2SRE estimates that adjust for publication bias. The T&F estimator has substantially lower variance than the PET-PEESE estimator, and the respective T&F-adjusted VSL estimates are $6.06 and $6.33 million. This large difference suggests that publication bias may be important, so we would include these estimates within the range of values to be considered for policy analysis. As noted earlier, 38 of the 49 estimates in Table 4 are between $6.0 and $10 million. Among the meta-regression estimates reported in Tables 5 and 7, the best fitting models are those with the 'datayear' variable included. However, the EPA Science Advisory Board recommended *not* controlling for the year of data collection in VSL meta-regressions citing a lack of a clear rationale for including it [51]. The specifications in Table 5 that exclude 'datayear' produce balanced VSL estimates between $8 and $10 million. These estimates fall safely within the central range of estimates from Table 4, so we would not expand that range based on the meta-regression specifications that do not control for publication bias or a time trend in Table 5. The best-fitting specifications among those that do control for publication bias using the PEESE estimator in Tables 7 and S2.7 also produce central VSL estimates within this range. Based on all of these considerations, we believe that a central estimate around $8 million and a range for sensitivity analysis between $6 and $10 million would be a reasonable overall synthesis of our results.

Another option would be to use a model averaging approach [56] to combine the results from multiple meta-regression models. To illustrate this approach, we applied jackknife model averaging (JMA)—which computes a weighted average of model outputs where the weights are chosen to minimize the sum of squared residuals of the resulting weighted average [57]—to 28 meta-estimates of the VSL in Tables 5–S2.7 (seven specifications × constrained or unconstrained $\sigma_{\mu,i}$'s × without and with correction for publication bias). The JMA weighted average of the estimated balanced VSL's (regression constants plus one half of the SP dummy coefficient) at the means of all control variables was $7.87 million.

A practical limitation of our application is that our meta-dataset includes only U.S. estimates of the VSL. For policy analyses in other countries, at least two options are available: (1) conduct a separate meta-analysis using domestic VSL estimates, or (2) adjust a U.S. VSL estimate to account for income differences between countries, as recommended by Viscusi and Masterman [58]. For instance, Lindhjem et al. [31] used meta-regression to analyze 856 VSL estimates from stated preference studies conducted in 38 countries. While a comprehensive reanalysis of Lindhjem's data is beyond the scope of this paper, we applied our 2SRE estimator to their meta-dataset in a side analysis. In Sect S3 of the Supporting information, we compare and contrast our results to the raw mean and the study-weighted mean VSL reported by Lindhjem et al. While to date there are limited or no data estimating the VSL in most low- and middle-income countries [59], our re-analysis of the Lindhjem data demonstrates that our estimation approach can be effectively applied to other meta-datasets of the VSL (or other effect sizes of interest) that include both point estimates and standard errors without practical barriers.

As a final caveat, we emphasize that the meta-estimates presented in this paper should not be construed as an official update of VSL values for use in U.S. EPA economic analyses or any other; rather, we present our application as a demonstration of the general estimation approach on realistic data. A natural next step would be to develop a more definitive

meta-dataset of VSL estimates to which a multilevel meta-analysis estimator like the 2SRE estimation approach developed here could be applied in a follow-up study.

## 5 Conclusions

In this study, we developed a two-stage random-effects meta-analysis estimator designed to handle unbalanced panels with one or multiple observations from each independent group of primary estimates. The method separates sampling from non-sampling sources of error within and between groups, leveraging the data to calculate efficient weights for clustered observations. We examined the performance of the estimator through a series of Monte Carlo experiments on constructed datasets. The simulation results demonstrate that the estimator performs well compared to several commonly used meta-analysis methods, including other multilevel estimators, particularly in scenarios with modest dataset sizes (20 to 60 studies, each contributing 1 to 15 observations) and significant heterogeneity within and between studies. At the tested sample sizes, the method is most effective when within-group non-sampling error variances are assumed to be homogeneous across groups. To illustrate the approach, we applied it to a meta-dataset comprising 113 value per statistical life (VSL) estimates derived from 10 revealed preference and 9 stated preference studies conducted in the United States between 1999 and 2019. Future meta-analyses in many domains—including but not limited to analyses of the VSL—may have similar data characteristics, so the usefulness of the 2SRE estimation approach or others like it should be correspondingly broad. Our findings suggest that wider adoption and further refinement of meta-analysis techniques, including multilevel estimators like the one developed here, can enhance the accuracy and precision of policy analyses. These advances are critical for synthesizing multiple estimates of policy-relevant metrics, including the VSL and many others.

## Supporting information

**S0 Graphical abstract.** Short supplemental abstract, largely visual.
(PDF)

**S1 Appendix.** Derivation of two-stage random-effects estimator.
(PDF)

**S2 Supplemental tables.** Monte Carlo simulation and meta-regression results.
(PDF)

**S3 Supplemental application.** Application to a global VSL meta-dataset.
(PDF)

**S4 Data and code.** Link to a Github repository containing data and code sufficient to reproduce our results.
(PDF)

## Acknowledgements

We are grateful for crucial insights from Mary Evans and Dan Phaneuf, detailed feedback from Charles Griffiths on an earlier draft, and wise counsel on general issues related to meta-analysis from Thomas Stanley and other members of the Meta-Analysis of Economics Research Network. All remaining conceptual and computational errors are our own. The findings and conclusions in this publication are those of the authors and should not be construed to represent any official USDA, US EPA, or U.S. Government determination or policy.

## Author contributions

**Conceptualization:** Stephen C. Newbold, Chris Dockins, Nathalie Simon, Kelly Maguire.

**Data curation:** Chris Dockins, Nathalie Simon, Kelly Maguire.

**Formal analysis:** Stephen C. Newbold, Abdullah Muhammad Sakib.

**Methodology:** Stephen C. Newbold, Chris Dockins, Nathalie Simon, Kelly Maguire, Abdullah Muhammad Sakib.

**Validation:** Abdullah Muhammad Sakib.

**Writing – original draft:** Stephen C. Newbold, Chris Dockins, Nathalie Simon, Kelly Maguire.

**Writing – review & editing:** Stephen C. Newbold, Chris Dockins, Nathalie Simon, Kelly Maguire, Abdullah Muhammad Sakib.

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
