## [Decision Letter · Decision Letter 0]

22 Dec 2024

PONE-D-24-25394A two-stage random-effects estimator for meta-analyses of the value per statistical lifePLOS ONE

Dear Dr. Newbold,

Thank you for submitting your manuscript to PLOS ONE. After careful consideration, we feel that it has merit but does not fully meet PLOS ONE’s publication criteria as it currently stands. Therefore, we invite you to submit a revised version of the manuscript that addresses the points raised during the review process.

**ACADEMIC EDITOR: Please respond carefully to reviewers comments.**

We look forward to receiving your revised manuscript.

Kind regards,

Ayman A. Swelum

Academic Editor

PLOS ONE

Journal Requirements:

2. As required by our policy on Data Availability, please ensure your manuscript or supplementary information includes the following: 

3. Please ensure that you refer to Figure 1 in your text as, if accepted, production will need this reference to link the reader to the figure.

4. Please remove your figures from within your manuscript file, leaving only the individual TIFF/EPS image files, uploaded separately. These will be automatically included in the reviewers’ PDF.

5. We notice that your supplementary [figures/tables] are included in the manuscript file. Please remove them and upload them with the file type 'Supporting Information'. Please ensure that each Supporting Information file has a legend listed in the manuscript after the references list.

Reviewers' comments:

Reviewer's Responses to Questions

**Comments to the Author**

1. Is the manuscript technically sound, and do the data support the conclusions?

Reviewer #1: Yes

Reviewer #2: Yes

2. Has the statistical analysis been performed appropriately and rigorously? 

Reviewer #1: Yes

Reviewer #2: Yes

3. Have the authors made all data underlying the findings in their manuscript fully available?

Reviewer #1: Yes

Reviewer #2: Yes

4. Is the manuscript presented in an intelligible fashion and written in standard English?

Reviewer #1: Yes

Reviewer #2: Yes

5. Review Comments to the Author

Reviewer #1: The manuscript titled “A two-stage random-effects estimator for meta-analyses of the value per statistical life” makes a commendable methodological contribution to the field of meta-analysis. The authors address significant challenges in synthesizing results from studies with unbalanced datasets while accounting for different sources of errors. The two-stage random-effects estimator proposed here is robust, rigorously tested using Monte Carlo simulations, and compares favorably against existing methods. The real-world demonstration using a VSL dataset adds practical weight to the study. However, a few refinements can further improve the clarity, accessibility, and impact of the manuscript.

The introduction section, while providing sufficient context, does not explicitly articulate the specific research gap that this study aims to address. While it is clear that the work focuses on improving precision in meta-analysis with unbalanced data, the novelty and objectives of the paper could be stated more clearly upfront. Explicitly stating the shortcomings of existing methods and how this approach resolves those issues will better highlight the importance of the work. Articulating these points will help readers appreciate the methodological advancements and the unique contributions of the proposed estimator.

The dataset used for demonstration, although well-documented and appropriate for testing the proposed method, is somewhat dated, spanning studies from 1999 to 2013 and limited to the U.S. context. While the methodological focus of the paper is clear, the reliance on older data may raise concerns regarding the broader relevance and applicability of the results. It would strengthen the paper if the authors explicitly acknowledged this limitation and discussed how applying the method to a more recent or globally diverse dataset might influence the findings. A short note suggesting this as future work would also enhance the study's scope.

The paper lacks a formal conclusion section, which leaves the manuscript feeling incomplete. While the results and discussion are thorough, a concise summary of the key contributions, findings, and implications would provide a stronger closure to the work. Including a brief conclusion that reiterates the main takeaways, such as the methodological advancements, the precision of the new estimator, and its comparative performance, would help solidify the impact of the study. Additionally, outlining areas for future research or potential applications of the estimator would add value for readers seeking next steps.

While the focus on methodological rigor is appreciated, the paper does not discuss practical applications or use cases where the proposed estimator could influence real-world decision-making. The study would benefit from a brief discussion of how more precise VSL estimates could improve cost-benefit analyses in areas such as public health, safety regulations, or environmental policy. Providing even a hypothetical example would help readers, especially policymakers or applied researchers, see the broader implications and practical value of the estimator. This would extend the paper’s relevance beyond academia and highlight its utility in shaping policy outcomes.

The technical nature of the paper, particularly in the methods and discussion sections, makes it inaccessible to non-specialist readers or students new to the subject. While the intended audience may be researchers and statisticians, simplifying parts of the discussion to include a plain-language summary of the findings would significantly improve accessibility. Briefly explaining the estimator’s benefits in non-technical terms and summarizing why it matters for meta-analysis would broaden the paper’s appeal to students, applied researchers, and practitioners.

The publication bias results, although tested with Trim-and-Fill and PET-PEESE methods, appear underemphasized in the paper. Since publication bias is a key concern in meta-analyses, summarizing these findings more clearly and discussing their implications for VSL estimates would provide additional insight. A short paragraph within the results or discussion section addressing this point would suffice.

Finally, while the figures and tables are helpful, some of them—such as the Monte Carlo simulation results—are dense and may be challenging for readers to interpret. Enhancing the figures with clearer labels, brief annotations, or takeaway messages would make the results more intuitive and engaging. Minor language refinements for flow and readability would also improve the overall presentation of the paper.

In conclusion, the authors have made a significant methodological advancement in meta-analysis that is both rigorous and innovative. The study addresses long-standing limitations in unbalanced datasets and error decomposition, and the proposed two-stage random-effects estimator is robustly tested. With clearer articulation of the research gap, an updated conclusion, practical use case discussions, and improved accessibility for broader audiences, this paper has the potential to make an even greater impact.

Reviewer #2: Add diagrams to illustrate the method. This may help readers better understand the context and estimation steps.

Complement the article with an analysis of a more recent and global data set, including studies after 2020, especially estimates for the COVID-19 period.

The article focuses basically on the United States and does not discuss the applicability of the method in other countries.

The specific benefits should be emphasized that the new estimation method could bring when designing regulations such as health, environmental or road safety.

Expand your explanation of the differences between methods and provide recommendations for when to use each technique.

6. PLOS authors have the option to publish the peer review history of their article (what does this mean?). If published, this will include your full peer review and any attached files.

Reviewer #1: No

Reviewer #2: No

---

## [Author Response · Author response to Decision Letter 1]

6 Feb 2025

Our replies to both reviewers can be found in our "Response to reviewers" document.

---

## [Decision Letter · Decision Letter 1]

21 Mar 2025

PONE-D-24-25394R1A two-stage random-effects estimator for meta-analyses of the value per statistical lifePLOS ONE

Dear Dr. Newbold,

Thank you for submitting your manuscript to PLOS ONE. After careful consideration, we feel that it has merit but does not fully meet PLOS ONE’s publication criteria as it currently stands. Therefore, we invite you to submit a revised version of the manuscript that addresses the points raised during the review process.

**ACADEMIC EDITOR: Please respond carefully for reviewer comments.**

We look forward to receiving your revised manuscript.

Kind regards,

Ayman A Swelum

Academic Editor

PLOS ONE

Reviewers' comments:

Reviewer's Responses to Questions

**Comments to the Author**

1. If the authors have adequately addressed your comments raised in a previous round of review and you feel that this manuscript is now acceptable for publication, you may indicate that here to bypass the “Comments to the Author” section, enter your conflict of interest statement in the “Confidential to Editor” section, and submit your "Accept" recommendation.

Reviewer #1: All comments have been addressed

Reviewer #2: All comments have been addressed

Reviewer #3: All comments have been addressed

2. Is the manuscript technically sound, and do the data support the conclusions?

Reviewer #1: Yes

Reviewer #2: Yes

Reviewer #3: Partly

3. Has the statistical analysis been performed appropriately and rigorously? 

Reviewer #1: Yes

Reviewer #2: Yes

Reviewer #3: Yes

4. Have the authors made all data underlying the findings in their manuscript fully available?

Reviewer #1: Yes

Reviewer #2: Yes

Reviewer #3: Yes

5. Is the manuscript presented in an intelligible fashion and written in standard English?

Reviewer #1: Yes

Reviewer #2: Yes

Reviewer #3: Yes

6. Review Comments to the Author

**Reviewer #1: **Thank you for your thorough revisions and thoughtful responses to the reviewer comments. The manuscript has significantly improved in clarity, structure, and methodological rigor. The research gap and novelty are now well-articulated, the discussion has been expanded to provide stronger comparisons with past studies, and the addition of a conclusion section has strengthened the overall presentation. The acknowledgment of dataset limitations and the refinement of methodological explanations further enhance the study's credibility. While minor areas such as dataset freshness and additional real-world applications could still be refined, they do not impact the core validity of the study. Given these improvements, I recommend accepting the manuscript with no further major revisions.

**Reviewer #2:** The authors have taken note of the observations and incorporated them into the article. I recommend the study for publication in this form.

**Reviewer #3:** The authors aim in their work to develop and present "A two-stage random-effects estimator for meta-analyses of

the value per statistical life".

Overall, the work was well done and presented comprehensively.

A brief comment: on line 160, there is a double "and"... it should simply be "and 3)...".

I also suggest that the authors provide a graphical abstract to better understand the manuscript.

However, how confident can the authors be that their new estimator performs similarly when applied to data from sources other than the United States? Is this performance similar to or significantly better than that of known estimators? It would be helpful for the authors to validate their methodology by applying this method to a database from a country similar to the United States and from a much more distant country (a developing country).

To answer my question, the authors have the choice between applying their method to two additional datasets for validation purposes (one relating to a European country and the other to a developed country) or to enrich the discussion in their article. I much prefer the first option.

7. PLOS authors have the option to publish the peer review history of their article (what does this mean?). If published, this will include your full peer review and any attached files.

Reviewer #1: No

Reviewer #2: No

Reviewer #3: No

---

## [Author Response · Author response to Decision Letter 2]

7 Apr 2025

Please see our response to reviewers pdf document.

---

## [Decision Letter · Decision Letter 2]

30 Apr 2025

A two-stage random-effects estimator for meta-analyses of the value per statistical life

PONE-D-24-25394R2

Dear Dr. Newbold,

We’re pleased to inform you that your manuscript has been judged scientifically suitable for publication and will be formally accepted for publication once it meets all outstanding technical requirements.

Kind regards,

Ayman A Swelum

Academic Editor

PLOS ONE

Additional Editor Comments (optional):

Reviewers' comments:

Reviewer's Responses to Questions

**Comments to the Author**

1. If the authors have adequately addressed your comments raised in a previous round of review and you feel that this manuscript is now acceptable for publication, you may indicate that here to bypass the “Comments to the Author” section, enter your conflict of interest statement in the “Confidential to Editor” section, and submit your "Accept" recommendation.

Reviewer #3: All comments have been addressed

2. Is the manuscript technically sound, and do the data support the conclusions?

Reviewer #3: Yes

3. Has the statistical analysis been performed appropriately and rigorously? 

Reviewer #3: Yes

4. Have the authors made all data underlying the findings in their manuscript fully available?

Reviewer #3: Yes

5. Is the manuscript presented in an intelligible fashion and written in standard English?

Reviewer #3: Yes

6. Review Comments to the Author

Reviewer #3: Thank you for your thoughtful revisions to the manuscript titled "A two-stage random-effects estimator for meta-analyses of the value per statistical life". I appreciate the attention to detail you have given to the feedback, and I can see the effort you have put into addressing the points raised.

The changes that the authors made have significantly improved the clarity and strength of the work, and I believe it is now in a much stronger position for publication.

7. PLOS authors have the option to publish the peer review history of their article (what does this mean?). If published, this will include your full peer review and any attached files.

Reviewer #3: No

---

## [Editor Report · Acceptance letter]

PONE-D-24-25394R2

PLOS ONE

Dear Dr. Newbold,

I'm pleased to inform you that your manuscript has been deemed suitable for publication in PLOS ONE. Congratulations! Your manuscript is now being handed over to our production team.

Kind regards,

on behalf of

Professor Ayman A Swelum

Academic Editor

PLOS ONE